# *RtcB2-PrfH* Operon Protects *E. coli ATCC25922* Strain from Colicin E3 Toxin

**DOI:** 10.3390/ijms23126453

**Published:** 2022-06-09

**Authors:** Tinashe P. Maviza, Anastasiia S. Zarechenskaia, Nadezhda R. Burmistrova, Andrey S. Tchoub, Olga A. Dontsova, Petr V. Sergiev, Ilya A. Osterman

**Affiliations:** 1Center of Life Sciences, Skolkovo Institute of Science and Technology, Moscow 121205, Russia; tinashe.maviza@skoltech.ru (T.P.M.); a.zarechenskaya@skoltech.ru (A.S.Z.); o.dontsova@skoltech.ru (O.A.D.); p.sergiev@skoltech.ru (P.V.S.); 2Department of Chemistry, Faculty of Bioengineering and Bioinformatics and Belozersky Institute of Physico-Chemical Biology, Lomonosov Moscow State University, Moscow 119992, Russia; nadya.b187@gmail.com (N.R.B.); andrei.chub@chemistry.msu.ru (A.S.T.); 3Shemyakin-Ovchinnikov Institute of Bioorganic Chemistry, Russian Academy of Sciences, Moscow 119992, Russia; 4Genetics and Life Sciences Research Center, Sirius University of Science and Technology, 1 Olympic Ave., Sochi 354340, Russia

**Keywords:** translation, bacteria, ribosome rescue, rRNA cleavage, RNA ligation, translation termination factor

## Abstract

In the bid to survive and thrive in an environmental setting, bacterial species constantly interact and compete for resources and space in the microbial ecosystem. Thus, they have adapted to use various antibiotics and toxins to fight their rivals. Simultaneously, they have evolved an ability to withstand weapons that are directed against them. Several bacteria harbor colicinogenic plasmids which encode toxins that impair the translational apparatus. One of them, colicin E3 ribotoxin, mediates cleavage of the 16S rRNA in the decoding center of the ribosome. In order to thrive upon deployment of such ribotoxins, competing bacteria may have evolved counter-conflict mechanisms to prevent their demise. A recent study demonstrated the role of PrfH and the RtcB2 module in rescuing a damaged ribosome and the subsequent re-ligation of the cleaved 16S rRNA by colicin E3 in vitro. The *rtcB2*-*prfH* genes coexist as gene neighbors in an operon that is sporadically spread among different bacteria. In the current study, we report that the RtcB2-PrfH module confers resistance to colicin E3 toxicity in *E. coli ATCC25922* cells in vivo. We demonstrated that the viability of *E. coli ATCC25922* strain that is devoid of *rtcB2* and *prfH* genes is impaired upon action of colicin E3, in contrast to the parental strain which has intact *rtcB2* and *prfH* genes. Complementation of the *rtcB2* and *prfH* gene knockout with a high copy number-plasmid (encoding either *rtcB2* alone or both *rtcB2*-*prfH* operon) restored resistance to colicin E3. These results highlight a counter-conflict system that may have evolved to thwart colicin E3 activity.

## 1. Introduction

A stunningly rich microbial community is vastly prevalent in most natural environments. Scarcity of resources and push for space are some of the essential determinants that shape the bacterial ecosystem. Bacteria have evolved an arsenal of tools to thwart potential contenders from colonization. Potent tool-box systems that bacteria exploit may either harm, kill or inhibit menacing competitors [1]. The acquisition of weapons through horizontal gene transfer is a common mechanism that bacteria utilize in order to assimilate and express small pieces of DNA that are raided from other microbes [2].

Nutrient deprivation in a growing population and genotoxic environmental stresses are some of the factors that are predicated to influence the evolution of a diverse arsenal of weaponry among bacterial communities, which directly modulates a broad spectrum of microbes [3,4,5]. In order to impair the proliferation of rivals, bacteria employ a diverse range of noxious agents that include not only small molecules, such as antibiotics, but also toxic proteins. Their mechanisms of action(s) range from interfering with cellular metabolic functions to the damage of essential components, such as the cell-wall envelope and associated protein cytoskeleton [6]. 

The translational machinery is one of the most popular targets for both toxins acting on neighboring cells, as well as intracellular toxins from toxin–antitoxin pairs. RelE cleaves mRNA in A-site [7,8]; some toxins have been characterized to cleave tRNAs, and other potent toxins are also known to damage the ribosome, with biochemical and structural evidence which demonstrates their ability to impair the 16S and the 23S rRNA, for example, colicin E3 and VapC20 toxins, respectively, among others [9,10,11]. Usually, there are specific restoration systems which could repair translation apparatus after toxin activity: tmRNA-SmpB could release ribosomes that are stalled on the cleaved mRNA [12,13,14,15] and RNA ligases could repair tRNA [16], but no apparent mechanism has been clearly demonstrated to render resistance to the action of ribotoxins in vivo. 

The most common catalytic active site among various RNA-targeting effectors is the RNase domain. Several substrates are generated by various mechanisms that are executed mainly by the metal-dependent and independent RNAse activity. The former generates two substrates with a 2′,3′-cyclic-PO4 on the 3′ end of the 5′ fragment and a 5′-OH at the 5′ terminus of the 3′ fragment via a transesterification reaction. The latter yields a 5′-PO4 on the 5′ end of the 3′ fragment and a 3′-OH at the 3′ end of 5′ fragment. In addition, other RNases, such as RloC further exert an exonucleolytic cleavage on the digested products, even though most of the RNA-targeting effectors exert endonuclease activity [17]. Deployment of protein effectors that target RNA generates a pool of diverse substrates, potentiating the need of an evolved RNA repair system for protection [18].

In order to catalyze ligation activity, classical RNA ligases join 5′-PO4 and 3′-OH termini in an ATP/Mg(II) dependent fashion. On the contrary, an atypical RNA ligase that includes RtcB mends a 2′,3′-cyclic-PO4 and 5′-OH RNA cleaved substrates, whose catalysis is GTP/Mn(II) dependent [19]. RtcB homologs are widely prevalent among a diverse pool of bacteria and several other species [18,20]. The solved crystal structures of RtcB homologs exhibit relatively similar structural configurations that are characterized by two core β-pleated sheets with nearly identical α/β fold secondary structures [18,21]. The two divalent manganese cation cofactors are nested between β-pleated sheets in the active site vicinity [18]. 

Upon cleavage by specific RNA-targeting effectors such as MazF, generated substrates that are specific to RtcB, 2′,3′-cyclic-PO4 and a 5′-OH, can be healed and sealed to retain functionality [22]. The mechanism of ligation that is mediated by RtcB is a three-step nucleotidyl transfer process summarized as follows: (i) The formation of a covalent RtcB-histidinyl-N-GMP intermediate upon the reaction of RtcB and GTP that is initiated by a conserved histidine moiety from the active site of RtcB which attacks the α-β phosphate linkage of an energy-rich nucleotide, GTP, a step that subsequently releases inorganic pyrophosphate. (ii) Coordinated by one of the accessible and active Mn^2+^ cofactors, the water molecule hydrolyzes the cyclic 2′,3′-PO4 3′ ends of a 5′ RNA fragment, wherein the GMP moiety is transferred to the freed 3′-PO4 from histidine. (iii) Lastly, a nucleophilic attack by the 5′-OH from the 3′ RNA fragment on the activated 3′-PO4 results in the formation of a phosphodiester bond linkage, a step which releases GMP [19,23,24,25]. The ligation catalysis by RtcB expands well beyond RNA to DNA level, wherein it ligates the OHDNA strands from damaged DNA through a direct transfer of GMP to the 3′-PO4 ends, resulting in the formation of a stable DNAppG structure. This resultant effect aids in protection from exonucleolytic degradation, wherein the generated structure acts as a primer for DNA synthesis by repairing polymerase machinery [26,27,28].

The bona fide RtcB from the RtcBA operon has been shown to elicit a protective effect in some species [16]. In *E. coli* for instance, the endo-nucleolytically cleaved 16S rRNA products by MazF are substrates for ligation by RtcB [22]. Furthermore, even complementation by *E. coli* RtcB in yeast *Δtrl1* was demonstrated biochemically to render a protective effect from fungal ribotoxins that target tRNA, including its ability to mediate mRNA splicing. RtcB homologs may have evolved to possess multifactorial functionality [29]. Strikingly, the RtcA protein (RNA cyclase) converts RNA products that cannot be recognized by RtcB from 3′-PO4 to potential 2′,3′-cyclic-PO4 ends that can be recognized for ligation activity [29,30,31].

Mechanistically, colicin E3 cleaves 16S rRNA in the decoding center of the small subunit, precisely between A1493 and G1494. The cleavage produces 2′,3′-cyclic-PO4 and 5′-OH substrates [9]. The overall consequence is an impaired translation, which eventually causes cell demise [11,18]. Alternative translation termination factors may contribute to bacterial antitoxin defense [20,32]. It was recently shown in vitro that colicin E3-damaged ribosomes are substrates for PrfH, which enters the A-site cavity of the ribosome, triggering peptidyl-tRNA hydrolysis and a subsequent split of the cleaved 70S ribosomes. After ribosomal subunits dissociation, the 30S small subunit is repaired by an RtcB2 protein, which re-seals the 2′,3′-cyclic-PO4 and a 5′-OH phosphate break [19,23,25]. In order to complement this result, the current study sheds light on the effect of colicin E3 in a cell-based system (in vivo) that this RtcB2-PrfH module elicits, as a potential viable counter-conflict system.

## 2. Results

### 2.1. RtcB2-PrfH Operon Protects E. coli ATCC25922 Strain from Colicin E3 Toxin

Recently, it was shown that the PrfH protein from *E. coli ATCC25922* could release fMet from fMet-tRNA bound to 70S ribosome that was cleaved by colicin E3 toxin, whilst the RtcB2 protein was further demonstrated to ligate damaged 16S rRNA substrate [19,23,24,33,34]. In order to assess the possible activity of these proteins in vivo, a set of single and double knockouts of *rtcB2* and *prfH* genes were generated in both *E. coli ATCC25922* and *BW25113* strains (Figure 1A). Wild-type and knockout genotypes were exposed to colicin E3 action in two different fashions: (i) the inducible expression of the toxin (Figure 1B,C); (ii) the addition of a recombinant toxin in vivo (Figure 1D–G). In both approaches that were performed, the results were generally similar: the cells of *E. coli ATCC25922* wild-type strain demonstrated resistance to colicin E3 activity, in contrast to the *E. coli BW25113* counterpart, which is sensitive to this toxin. Deletion of *rtcB2*, *prfH* or both genes in *E. coli ATCC25922* led to a significant increase in sensitivity to colicin E3 toxicity. Single and double knockouts in *E. coli BW25113* did not demonstrate any plausible difference relative to the wild-type background, emphasizing that the RtcB2-PrfH module is not functional in this strain due to the truncation of the operon (Figure 1A).

To prove the protective effect of RtcB2 and PrfH against colicin E3 toxic effect, complementation experiments were conducted. *E. coli BW25113* and *ATCC25922* strains carrying the knockout of entire *rtcB2*-*prfH* operons were transformed with the plasmids encoding either *rtcB2*, *prfH* or both *rtcB2*-*prfH* genes. Hyper-expression of both genes (*rtcB2* and *prfH*) or *rtcB2* alone protected both strains from the toxic effect that was caused by colicin E3, while the strains harboring plasmid with the *prfH* gene only remained sensitive to the ribotoxin (Figure 1F,G). The hyper-expression of both genes was demonstrated by a qPCR (Appendix A).

### 2.2. Cleavage Fragments Generated by Colicin E3 Were Detected in Total RNA Purified from Treated Cells

The colicin E3 protein is a well characterized rRNAse that cleaves 16S rRNA in the decoding center, yielding products of about 50 and 1500 nucleotide-long fragments [9,11]. In order to assay for this cleavage, log-phase cells were incubated for 2 h at 37 °C with colicin E3, subsequently followed by total RNA purification and denaturing gel electrophoresis. The RNA that was purified from untreated cells was used as a negative control.

Treatment by colicin E3 induced cleavage of the 16S rRNA for both wild-type and knockout genotypes for *E. coli BW25113* (Figure 2A and Appendix A). On the contrary, *E. coli ATCC25922* wild-type strain, as well as both *prfH* and *rtcB2* single gene knockouts exhibited a low cleavage signal upon colicin E3 treatment, while the double knockout demonstrated an efficient 16S rRNA cleavage (Figure 2B and Appendix A). Taken together, this result further complements the survival assays (Figure 1B,C). Particularly for the *E. coli ATCC25922* wild-type, a strong protective effect was aided by the RtcB2-PrfH module which clearly demonstrated the importance of the retained intact genes, in contrast to the *E. coli BW25113* counterpart.

Complementation of the double knockout by a plasmid carrying both *rtcB2*-*prfH* genes from the *E. coli ATCC25922* strain provided a strong protective effect in the presence of colicin E3 as observed in Figure 2C,D, as expected. In summary, the synergistic effect of RtcB2 and PrfH proteins helps to alleviate colicin E3 toxicity (Appendix A).

### 2.3. Primer Extension Analysis of the 16S rRNA Integrity in the Presence of Colicin E3, PrfH and RtcB2 In Vivo

The primer extension assay was utilized to score for the exact site of colicin E3 cleavage. Colicin E3 cleaves the 16S rRNA between the nucleotides A1493 and G1494. Both wild-type and knockout *E. coli BW25113* and *ATCC25922* strains suffer extensive cleavage of the 16S rRNA upon colicin E3 treatment (Figure 3A,B and Appendix A), relative to untreated cells. The *E. coli ATCC25922* wild-type strain seems to exhibit a high intense cDNA abundance corresponding to colicin E3 treatment. This result is unexpected, since the intact RtcB2-PrfH module is present in the genome of *E. coli ATCC25922* strain and protects bacteria from colicin E3.

Complementation of the *rtcB2*-*prfH* knockout genotype by a plasmid bearing both *rtcB*-*prfH* genes demonstrated synergistic protection from colicin E3 cleavage (Figure 3C,D and Appendix A). This result corroborates strongly with our previous results (Figure 1F,G and Figure 2C,D). Indeed, when ribosomes are blocked after colicin E3 treatment, PrfH seems to help with the disassembly of the damaged ribosome, aiding in the subsequent ligation of the cleaved 16S rRNA by RtcB2. Expression of the *rtcB2* gene alone does seem to decrease the steady-state level of cleaved 16S rRNA upon continuous exposure to colicin E3. This is somewhat contrasting to our observation that *rtcB2* overexpression is sufficient to restore cell viability upon colicin E3 treatment even without a functional *prfH* gene.

Apart from the expected primer extension stop at the 16S rRNA nucleotide G1494 upon colicin E3 treatment, we occasionally observe additional reverse transcriptase stops, such as, at the nucleotide C1496 in the *E. coli ATCC25922* strain (Figure 3B). Although we have no straightforward explanation for this observation, we might hypothesize that it is due to exo- or endonucleolytic 16S rRNA degradation which is secondary to primary cleavage by colicin E3.

### 2.4. RtcB2-PrfH Operon Does Not Protect E. coli Cells from the tRNA Targeting Toxins

A plethora of toxins are known to cleave several components in the cell which are essential for cell viability. Having shown the potential substrate of the RtcB-PrfH module for protection from colicin E3 activity towards the 16S rRNA, we decided to check and determine whether this module aids in protection against the cleavage of tRNAs by the corresponding toxins in vivo. In order to demonstrate this, we used *E. coli* strains, *BZB2108* and *BZB2103*, that host the plasmid expressing colicin E5 and D proteins in the periplasmic space, respectively [35,36,37]. The protein that was excreted into the media was clarified from the cell pellet, concentrated, filter sterilized and used for the survival assays. Biochemical evidence shows that colicin E5 preferentially cleaves tRNA-Tyr, tRNA-His, tRNA-Asn and tRNA-Asp, whereas colicin D cleaves the anticodon loop of tRNA-Arg [38,39].

Colicin E5 and D produce 2′,3′ cyclophosphate and 5′-OH RNA ends, similar to colicin E3, and one would expect a protective action from the RtcB2-PrfH module for these colicins [38,40,41,42,43,44,45]. The survival assay results that are shown in Figure 4 demonstrate that the RtcB2-PrfH module fails to protect bacteria from tRNA-targeting toxins, colicin E5 and D. This reinforces the idea that the RtcB2-PrfH module has evolved specifically to protect cells from colicin E3 toxicity.

## 3. Discussion

In a diverse pool of bacterial communities, some of the well characterized and sporadically spread proteins with a strong similarity to bacterial release factors include ArfB and PrfH. Structurally, these two peculiar proteins retain a highly conserved GGQ motif. The ArfB protein has an extended C-terminal α-helical region which is essential for recognizing and scanning ribosomes that are jammed on non-stop complexes [15,20,32,46]. In contrast to ArfB, PrfH exhibits an anticodon recognition domain. Relative to canonical release factors, PrfH lacks an N-terminal GTPase-interacting domain. The *prfH* gene is typically harbored in an operon that codes for an RtcB2 protein [18,20,32]. The RtcB2-PrfH module is implicated to function as a counter-conflict system that mediates RNA repair in biological warfare that is potentially triggered by environmental stress. This system may have evolved to protect RNA from deployed toxin effectors, whose target may also include the impairment of translational apparatus among competing bacteria, in a microbial ecosystem [20]. In the current study, we demonstrate for the first time the rescue of colicin E3 damaged ribosomes by a counter-conflict system from the RtcB2-PrfH module in vivo. The basis of our current results is a highlight from two *E. coli* strains, namely *BW25113* and *ATCC25922* strains. The former harbors a truncated operon, whilst the latter exhibits a full-length intact operon [20,32,47].

Single and double knockout genotypes of *prfH* and *rtcB2* were generated in order to search for the conditions that impair the phenotypic effect under specific stress conditions. Colicin E3 treatment significantly impaired growth in knockout genotypes. In particular, the induction of colicin E3 in vivo obliterates bacterial growth on knockout strains of the *E. coli ATCC25922* strain, in contrast to the wild-type background, whereas *BW25113* strains, which harbors truncated genes from the rescue module suffers an impaired growth defect both on wild-type and knockout genotypes. Taken together, the primary results led us to speculate that the expression of *rtcB2* and *prfH* in a wild-type strain from *E. coli ATCC25922* significantly impacts survival in the presence of colicin E3 stress, due to the expression of functionally intact active proteins, in contrast to the *E. coli BW25113* counterpart.

We then asked whether complementation by single or both *rtcB2* and/or *prfH* genes aid in rescue and protection under colicin E3 treatment. The expression of both genes drastically renders resistance to colicin E3 toxicity. Interestingly, the expression of *rtcB2* alone provided a strong protective effect from the action of the rRNAse from our survival assay. This makes us speculate that the *prfH* gene can be dispensable when the *rtcB2* gene is expressed at high levels alone. Colicin E3 may cleave the 30S small subunit prior to the formation of a competent 70S ribosome at initiation stage, which may necessitate the need for repair by RtcB2, an independent event that does not require PrfH. In addition, alternative ribosome disassembly factors may aid in alleviating the colicin E3 damaged ribosomes when PrfH is absent, which still warrants further experimental validation, but this is beyond the scope of the current study. In the same vein, a single knockout genotype for *E. coli ATCC25922 ΔprfH* demonstrated a low growth defect when exposed to colicin E3. This may be a consequence of the different level of *rtcB2* expression between *ΔprfH* cells and in the *ΔrtcBΔprfH* strain transformed with pRtcB2 plasmid. The level of expression of *rtcB2* in the *ΔprfH* genotype may be insufficient to maintain a normal cell growth under toxin treatment. Taken together, PrfH might either aid RtcB2 activity, by attracting RtcB2 to the damaged ribosomes for instance, or it may somehow decrease the phenotypic cost of 16S rRNA cleavage, aided by the release of stalled ribosomes, thus enhancing cell survival even at a slower rate of 16S rRNA repair by RtcB2. Involvement of other ribosome rescue factors in an RtcB2-guided 16S rRNA repair in cells devoid of PrfH might also explain the colicin E3 resistance of bacteria upon overexpression of RtcB2 alone.

The results from the primer extension analysis speak in favor of the possibility that, after the initial colicin E3-mediated 16S rRNA cleavage, an additional degradation of the rRNA may follow. Thus, a pathway of rRNA damage and repair might seem more complex than is currently understood.

There is an existing bonafide *rtcB* gene from the RtcBA operon whose main role is universal in mediating the RNA repair of cleaved tRNA(s) and MazF cleaved substrates [16,22]. Though RtcB may recognize substrates that are similar to colicin E3 products, its gene does not render resistance to colicin E3 as it is harbored in the genome of both the wild-type and knockout strains that were used in this study. It is likely that several RtcB homologs may have been evolutionarily selected for specific targets that may aid in the protection against a plethora of toxins. In this work, we demonstrated the need for RtcB2 to ligate products of cleaved 16S rRNA substrates, with no apparent protection observed upon treatment by tRNA targeting toxins, colicin E5 and D, whose toxicity confers sensitivity to both double knockout genotypes in both strains, *E. coli BW25113* and *ATCC25922* strains, complemented with *rtcB2* and/or *prfH*. Our complementation assays under stress with tRNA targeting toxins revealed that the RtcB2-PrfH counter-conflict module may be dispensable for cell viability in bacteria. Though the RtcB superfamily of ligases exhibit similar folding and a strong degree of sequence similarity, they tend to differ in peculiar structural features, and this difference may necessitate the distinction of a plethora of substrates expanding from a pool rRNA, tRNA and mRNA.

Taken together, our results demonstrate a synergistic effect of the counter-conflict system, RtcB2-PrfH module, on its ability to potentiate a protective effect on damaged ribosomes by colicin E3. This preliminary study enables further perspectives in understanding this kind of defense system. Given the diversity of RNases acting upon the translation machinery, one could expect the existence of several corresponding defense armaments.

## 4. Materials and Methods

### 4.1. Generation of Knockouts Strains

Primers were designed for the deletion of either single or double knockouts of *prfH* or *rtcB2* (Appendix A), as described previously [48]. The initial PCR was performed using 2 ng of pKD4 plasmid as a template with a Q5^®^ High-Fidelity 2X Master Mix (New England Biolabs, Ipswich, MA, USA) to amplify the kanamycin cassette with FRT sites. The PCR product was run on an ethidium bromide-stained 1% agarose gel and then visualized under a UV transilluminator. The successful PCR product was then DpnI digested and PCR purified using a Qiagen PCR kit (Qiagen, Hilden, Germany). The concentration was then determined using a nanodrop.

Electrocompetent cells for *E. coli ATCC25922* and *BW25113* strains were transformed with pKD46 and grown for ~2 h at 30 °C in 10 mL LB. The transformants were spun at 3000 rpm for 5 min and a volume of ~200 µL was left in the Eppendorf tube. About 100 µL of the transformants were plated in LB-kanamycin plates and left to grow at 37 °C overnight. A colony PCR procedure was then performed using primer pairs that annealed to the kanamycin cassette and region(s) on the genomic DNA distal from the recombination site(s). An additional PCR was used to score the unique difference between the *E. coli ATCC25922* and *BW25113* strains (Appendix A). Positive PCR hits were then grown in LB media with kanamycin and streaked to single colonies. In order to check if pKD4 was successfully removed, the colonies for the scored knockout genotypes were further screened for ampicillin and kanamycin sensitivity (which were patched colonies on plates) in this order, and left to grow overnight at 37 °C. Cells that grew only in kanamycin and not in ampicillin LB-agar plates were then used to prepare overnight cultures, wherein glycerol stocks were further prepared and stored at −80 °C.

Excision of the kanamycin cassette was executed as previously described [48,49,50]. In order to remove the kanamycin cassette with FRT sites, electrocompetent cells were prepared for the knockouts that were generated and transformed with the pCP20 plasmid. The transformants were then grown at 30 °C in LB-ampicillin plates. A single colony was inoculated and grown in 5 mL LB broth media at 43 °C to induce the FLP recombinase. The overnight culture was then streaked to single colonies and grown at 30 °C on an LB agar plate. Several individual colonies were then screened on LB-kanamycin, LB-ampicillin and LB agar plates in this order by patching using sterile 200 µL tips. Positive clones that grew on LB agar only were then used to prepare overnight cultures wherein glycerol stocks were prepared and stored at −80 °C. Furthermore, an additional PCR check was carried out using primers that amplified from regions distal to the excised regions to confirm the successful hits (Appendix A).

The knockout strains that were sensitive to kanamycin were then used for preparing competent cells that were used for transformations with plasmids carrying either single or double genes encoding *prfH* and/or *rtcB2* from the *E. coli ATCC25922* genome, including a control plasmid that lacked either of these genes under study. The plasmids that were used for complementation were IPTG inducible, under a constitutive *T5* promoter, with a pET backbone conferring kanamycin resistance.

### 4.2. Protein Purification

The plasmid expressing colicin E3 toxin and immunity protein (Im3) antitoxin (with a His-tag located in the N-terminal end), was used to purify the rRNAse toxin under denaturing conditions (the plasmid was gifted by Professor Colin Kleanthous of the University of Oxford). The protein purification procedure was generally similar to the one previously described, with some minor modifications [51]. The expression plasmid was transformed into *E. coli* BL21(DE3) competent cells and plated on LB-ampicillin agar plates. An overnight culture of *E. coli* BL21(DE3) cells from the successful transformants with the expression plasmid were then grown in LB-ampicillin media. The saturated overnight culture was then diluted 1:100 in LB media containing ampicillin, supplemented with glucose (0.2% final concentration), and then grown at 37 °C until log-phase with shaking at 250 rpm until an optical density of ~0.5 was reached, which was tracked by measuring absorbance at an absorbance of 600 nm. The temperature was then dropped to 16 °C and the cells were further shaken for an additional 20 min. A sample before induction was collected, followed by the addition of isopropyl β- d-1-thiogalactopyranoside (IPTG) to a final concentration of 0.5 mM. This was left shaking overnight for about ~18 h at the same speed and the sample after induction was then collected. The minus and plus IPTG samples (100 µL) were spun down at maximum speed, followed by the complete removal of the supernatant, and then resuspended in HU buffer (200 mM Tris-HCl at a pH of 6.8, 8 M UREA, 5% SDS, 1 mM EDTA, with an addition of fresh dithiothreitol (DTT) to a final concentration of 100 mM), heated at 95 °C for 4 min and loaded on a 12% SDS-PAGE casted gel. The successful induction of colicin E3 was detected upon staining of the gel using InstantBlue^®^ Coomassie Protein Stain reagent (Sigma-Aldrich) with an expected molecular weight of ~60 kDa. The induced cells were then pelleted by centrifugation at 10,000 rpm for 30 min and successively lysed in a binding buffer (5 mM imidazole, 500 mM NaCl and 20 mM Tris-HCl with a pH of 7.5) that was supplemented with 0.3 mg/mL lysozyme and PMSF (to a final concentration of 0.5 mM). The cell pellet was then resuspended to homogeneity with the prepared lysis solution and subsequently incubated on ice for 30 min. The cells were then sonicated using the following regime: an amplitude of 80 with 30 s sonication and a break at 1-min intervals, repeated about 5 times. The samples were then further incubated on ice for 10 min, followed by centrifugation at 12,000 rpm for 30 min, executed twice. The lysate was then filtered through a 0.2 micron syringe filter and used for purification using the ÄKTA Purifier (GE Healthcare, Chicago, IL, USA). Briefly, the lysate was bound on an equilibrated Ni-NTA resin (cOmplete His-Tag Purification Column-Sigma-Aldrich) for at least 30 min, succeeded by collection of the flow-through (Roche, Mannheim, Germany). The binding buffer was used to wash the resin until a baseline equilibration at approximately zero was achieved, in order to eliminate non-specific binding proteins from resin whose absorbance was detected in-line at 280 nm. Furthermore, the binding buffer containing 60 mM guanidinium chloride was then utilized until the non-specific binding proteins were no longer detected at an absorbance of 280 nm inline. Another wash with binding buffer only was performed in a similar fashion, followed by elution under denaturing conditions with a binding buffer containing 6 M guanidinium chloride. The eluted protein was collected and checked on a 12% SDS-PAGE gel and then renatured extensively in a dialysis buffer (Milli-Q ultrapure water) with several switches to a new dialysis buffer at least ~4 times. The renatured protein was then checked again on a 12% SDS-PAGE gel (Appendix A), followed by measurement of the concentration using a nanodrop, which was ~2 mg/mL. The activity of colicin E3 was assayed in vitro using the S30 kit (Promega, Madison, WI, USA).

### 4.3. Complementation Assay

To obtain plasmids for complementation, each of the genes encoding *prfH*, *rtcB2* and *rtcB2*-*prfH* operons (from *E. coli ATCC25922*) without tags were cloned into a pET30a-tev vector with a T5 lac-promoter. The plasmids were named pPrfH, pRtcB2 and pRtcB2-PrfH, respectively. The pET30a-tev vector containing an extrinsic gene non-related to translational process was used as a negative control.

*E. coli* chemically competent cells of *BW25113 ΔrtcB2ΔprfH* and *ATCC25922 ΔrtcB2ΔprfH* strains were transformed with the complementation plasmids: pPrfH, pRtcB2, pRtcB2-PrfH and the control plasmid (pControl) using the chemical transformation method. Overnight cultures of these cells were diluted 1 to 100 and grown at 37 °C to early log phase. Induction of the genes of interest was due to promoter leakage. Recombinant colicin E3 protein was added to a final concentration of 1 mg/mL. Other samples were treated with an equal amount of Milli-Q ultrapure water as a negative control for the experiment. All samples were incubated at 37 °C for 2 h.

Afterwards, the cells were diluted to 0.01 and subjected to six ten-fold serial dilutions. A volume of 5 μL for each dilution was spotted on LB-agar plates, followed by incubation for 18 h at 37 °C. Three parallel experiments were carried out. The results were then viewed using the Image Lab software (Bio-Rad, Hercules, CA, USA).

### 4.4. In Vivo Testing of Colicin E3 Activity on Wild-Type and Knockout Strains after Inducible Expression

The gene-encoding colicin E3 was cloned into a pASK-IBA4 vector containing a tet-repressor under a tet-inducible promoter. The obtained plasmid was named pColE3. *E. coli* chemically competent cells of *BW25113* (wild-type, *ΔrtcB2*, *ΔprfH*, *ΔrtcB2-ΔprfH*) and *ATCC25922* (wild-type, *ΔrtcB2*, *ΔprfH*, *ΔrtcB2-ΔprfH*) strains were transformed with the plasmid pColE3 using the chemical transformation method. Overnight cultures of each cell were diluted 1 to 100 and grown at 37 °C to early log-phase. Anhydrotetracycline (200 µg/mL) was added to half of the samples in the ratio 1 to 1000 for colicin E3 induction. All samples were incubated at 37 °C for 2 h.

Afterwards, the cells were diluted to 0.01 and subjected to ten-fold serial dilutions. A volume of 5 μL for each dilution was spotted on LB-agar plates, followed by incubation for 18 h at 37 °C. Three parallel experiments were carried out. The results were then viewed using the Image Lab software (Bio-Rad).

### 4.5. In Vivo Testing of Recombinant Colicin E3 Activity on Wild-Type and Knockout Strains

Overnight cultures of *E. coli* cells of *BW25113* (wild-type, *ΔrtcB2*, *ΔprfH*, *ΔrtcB2-ΔprfH*) and *ATCC25922* (wild-type, *ΔrtcB2*, *ΔprfH*, *ΔrtcB2-ΔprfH*) strains were diluted 1 to 100 and grown at 37 °C to early log-phase. Recombinant colicin E3 protein was added to a final concentration of 1 mg/mL. Other samples were treated with an equal amount of Milli-Q ultrapure water as a negative control of the experiment. All samples were incubated at 37 °C for 2 h.

Afterwards, the cells were diluted to 0.01 and subjected to ten-fold serial dilutions. A volume of 5 μL for each dilution was spotted on LB-agar plates, followed by incubation for 18 h at 37 °C. Three parallel experiments were carried out. The results were then viewed using the Image Lab software (Bio-Rad).

### 4.6. RNA Purification

Total RNA was purified using the TRIzol™ Reagent, according to the manufacturer’s instructions (Ambion, Carlsbad, CA, USA). Briefly, the untreated and treated cells were pelleted, followed by the complete removal of LB. The cell pellet was resuspended in TRIzol™ reagent in order to allow the complete dissociation of nucleoprotein complexes. Chloroform (0.2 mL) was used for cell lysis and the samples were further incubated for 2 min at room temperature. Centrifugation of the samples was then performed for 15 min at 12,000× *g* at 4 °C. After the centrifugation procedure, the upper colorless layer was carefully transferred to a new sterile 1.5 mL Eppendorf tube. A volume of 0.5 mL isopropanol was then added to the upper-phase isolated sample in order to precipitate RNA, subsequently followed by a 10-min incubation period at 4 °C. The samples were then centrifuged at 12,000× *g* at 4 °C. The supernatant was carefully removed and trashed using a micropipette without disturbing the white gel-like pellet on the tube. The RNA was then carefully resuspended in 75% ethanol. The tubes were vortexed briefly and centrifuged for 5 min at 7500× *g* at 4 °C. The supernatant was completely removed and the tubes were air dried for 10 min. The RNA was then resuspended in Milli-Q ultrapure water. The RNA concentration was measured using a nanodrop.

### 4.7. RNA Gels

The isolated RNA (~500 ng) was used to run a pre-phoresed 20% denaturing polyacrylamide UREA gel electrophoresis. Firstly, the samples were aliquoted into PCR strips, followed by the addition of FA loading buffer. The samples were mixed, spun-down and heated at 95 °C for 4 min. A volume of 5 µL was loaded onto the prepared gel and ran under 1X TBE buffer. The gel was then stained with SYBR™ Green II RNA Gel Stain (ThermoFisher Scientific, Waltham, MA, USA) for 40 min while shaking under complete darkness. The gels were then scanned and visualized using the Image Lab software (Bio-Rad).

### 4.8. Primer Labeling

For a total volume of 20 µL, the following contents were added to the mixture: 4 µL of reverse primer (CCTACGGTTACCTTGTT) with an initial concentration of 20 µM; 2 µL of 10X reaction Buffer A (ThermoFisher Scientific, Waltham, MA, USA); 2 µL of labeled γ-[32P] ATP; 10.2 µL of Milli-Q ultrapure water; and 1.8 µL of T4 Polynucleotide Kinase (Thermo Fisher Scientific, Waltham, MA, USA). The samples were then mixed, spun down and incubated for 30 min at 37 °C, followed by storage at −20 °C.

### 4.9. Sequencing Reactions Preparation

The PCR product of 16S rRNA was amplified using pAM522 plasmid as a template with appropriate primers that would enable detection of the cleavage signal zone for the preparation of the sequencing reactions. The following protocol was utilized for PCR amplification using a Q5^®^ High-Fidelity 2X master mix (NEB, Ipswich, MA, USA); 98 °C-30 s, 98 °C-10 s, 59 °C-30 s and 72 °C-10 s for 30 cycles, followed by a final extension at 72 °C-30 s and storage at 12 °C. The PCR product was analyzed on a 1% gel and then purified using a Qiagen PCR kit, according to manufacturer’s protocol (Qiagen, Hilden, Germany). The concentration was determined using a nanodrop, thereafter.

For the sequencing reactions, 4 µL of ddGTP, ddATP, ddTTP and ddCTP provided by the manufacturer were placed into separate PCR tubes (ThermoFisher Scientific, Vilnius, Lithuania). All the preparatory reactions were performed under ice conditions. A mastermix with 1 µL template DNA (PCR product); 11.5 µL Milli-Q ultrapure water; 2 µL of reaction buffer; 1 µL of labeled primer; and 2 µL of DNA polymerase were added, mixed, spun-down and aliquoted (4 µL) in to the PCR tubes with ddNTPs. The samples were spun down briefly, followed by the addition of 4 µL of mineral oil. Upon spinning down the samples, the reactions were then placed in a thermocycler at 95 °C-2 min, 95 °C-30 s, 55 °C-30 s and 72 °C-1 min for 40 cycles, followed by a final extension at 72 °C-2 min and storage at 4 °C. A stop solution (8 µL) was then added to the sequencing reactions and stored at −20 °C until needed.

### 4.10. Primer Extension

A concentration of ~500 ng RNA was used as a template for the primer extension. First, annealing of the primer to the RNA was performed using the following cocktail (for 1 reaction): 500 ng of RNA (2.5 µL); 0.25 µL of 2 µM labeled primer; 0.25 µL of 10 µM dNTPs; and 4.25 µL of MilliQ ultrapure water. A non-template sample (MilliQ ultrapure water) was added as a negative control of the experiment. The samples were mixed, spun down and incubated at 85 °C for 1 min, followed by a 1 °C temperature drop to 26 °C over a 30 min window. The samples were then placed on ice for 2 min.

The second stage was cDNA synthesis using *Avian myeloblastosis* virus (AMV) reverse transcriptase (Roche, Mannheim, Germany). The reaction components for one reaction included: 2 µL AMV buffer (Roche, Mannheim, Germany), 0.25 µL RiboLock RNase Inhibitor (ThermoFisher Scientific, Waltham, MA, USA) and 0.5 µL AMV reverse transcriptase (Roche, Mannheim, Germany). The samples were carefully mixed, spun down and incubated at 42 °C for 30 min for cDNA synthesis. The reaction was then stopped by the addition of 10 µL formamide solution (a 1 mL stock with 980 µL formamide (FA) (deionized, nuclease free, Ambion)), 20 µL of 0.5 M EDTA (pH8.0), 1 mg of bromophenol blue and 1 mg of xylene cyanol). The samples were then heated for 4 min at 95 °C and 3 µL were loaded onto a pre-phoresed 20% polyacrylamide gel. For a 40 mL volume preparation of gel the following components were mixed: 4 mL of 10X TBE buffer; 10 mL 40% acrylamide/bisacrylamide; 19.2 g (8 M final concentration) UREA; and then finally topped to the 40 mL volume mark with deionized water, followed the by addition of 40 µL TEMED and 400 µL of 10% (*w*/*v*) APS) for gel casting, under 1X TBE running buffer. The same volume of sequencing reactions was also added to the casted gel, which was followed by running the gel at 10 W for 30 min, then further increased to 37 W until the xylene cyanol band was located towards the end of the gel. The gel was then transferred to a filter paper, dried, and transferred to a cassette, then covered with film and left overnight for signal development. The film was then scanned at 750–1000 V using Typhoon to detect the cDNA abundance from the P-32 signal.

### 4.11. Crude Isolation of Colicin E5 and D

*E. coli* host strains (*BZB2108* and *BZB2103*) bearing colicin E5 and D plasmids express and secrete the toxins into the periplasmic space [37]. The host strains were plated from stocks in LB-agar plates that contained nalidixic acid. Each strain was grown overnight in 100 mL LB media. The saturated culture was then centrifuged at maximum speed for 20 min, followed by transfer of the supernatant into sterile 50 mL falcon tubes. This procedure was performed twice to remove residual debris. The supernatant was then concentrated using an Amicon^®^ Ultra-15 Centrifugal Filter Unit with a Nominal Molecular Weight Limit (NMWL) of 3000 Da (Millipore, Cork, Ireland). The concentrate was then filtered through a 0.2-micron unit and used for survival assays.

### 4.12. Expression Profile Analysis Using qRT-PCR

The first stage for the preparation of the qRT-PCR was the removal of genomic DNA from 1 μg total RNA that was purified using the DNase I, RNase-free kit in accordance with the manufacturer’s protocol (ThermoFisher Scientific, Waltham, MA, USA). Briefly, the cocktail for a single reaction included: RNA (1 μg); 10X reaction buffer (1 μL); DNase I, RNase-free (1 μL (1U)); and DEPC-treated water (added to 10 μL). The samples were incubated at 37 °C for 30 min. In order to prevent RNA degradation, a volume of 1 µL of 50 mM EDTA chelating agent was then added to the mixture with a further incubation at 65 °C for 10 min. The prepared RNA was used as a template for cDNA synthesis using the Maxima First Strand cDNA Synthesis Kit for qRT-PCR (ThermoFisher Scientific, Waltham, MA, USA). The components for reverse transcription for a single reaction included: 5X Reaction Mix (4 µL), Maxima Enzyme Mix (2 µL), template RNA (2.5 μL), and nuclease-free water added to 20 µL. The samples were then incubated for 10 min at 25 °C which was then followed by a further incubation window of 15 min at 50 °C. The reaction was terminated by heating at 85 °C for 5 min. The generated cDNA that was synthesized was used as a template for the qRT-PCR, wherein the Maxima SYBR Green qPCR Master Mix (2X) kit was utilized for the analysis (ThermoFisher Scientific, Waltham, MA, USA). The contents for a single reaction had the following contents: Maxima SYBR Green qPCR Master Mix (2X) (10 μL); forward and reverse primers (1 μL from a 5 μM primer-mix stock) (Appendix A); and nuclease-free water (19 μL). Furthermore, 10 μL of mineral oil was added to the reaction mixture. The samples were then spun-down and loaded onto the qRT-PCR thermocycler under SYBR plate-scan protocol using the following conditions: 95 °C-10 min; 95 °C-15 s, 56 °C-30 s, 7 °C-30 s for 40 cycles; 95 °C-10 s, 55 °C-10 min, 95 °C; and then lastly left idle at 12 °C at the end of the run. The set of primers used were elongation factor G (housekeeping gene), *prfH* and *rtcB2*-specific primers that amplify ~100 bp of the respective target gene. The reactions were executed with technical replicas in triplicate. The calculations were scored using Microsoft Excel, wherein the difference between the housekeeping gene and the target gene was first scored (∆C_q_). Then, the ∆C_q_ expression was deduced by using the following expression, 2^−∆Cq^. The mean ∆C_q_ expression and standard deviation were calculated across all replicas using default formulas from Microsoft Excel. Thereafter, the values for the treated samples scored were then divided by the untreated counterpart, in order to calculate ∆∆C_q_ expression. The ∆∆C_q_ for standard deviation was also performed in a similar fashion. The relative quantity was calculated in contrast to pControl for the *E. coli ATCC25922 ∆rtcB2∆prfH* genotype with plasmids bearing either *prfH* and/or *rtcB2*, used for complementation assay. A bar graph plot was generated along with its corresponding deciphered standard deviations.

## 5. Conclusions

The current study is an extension of a previous in vitro demonstration study of the RtcB2-PrfH module in the rescue and repair of colicin E3-damaged ribosomes. We further complement the in vitro study using a cell-based system, wherein we show the indispensable effect that is exerted by this module as an essential feature for cell viability under stress by colicin E3. We additionally show that the expression of the *rtcB2* gene alone seems to render resistance to the ribotoxicity that is caused by colicin E3. This still warrants further investigation on its possible reparation role in cleaved 16S rRNA, mediated either alone or with assistance by other factors when PrfH is absent.

## Figures and Tables

**Figure 1 ijms-23-06453-f001:**
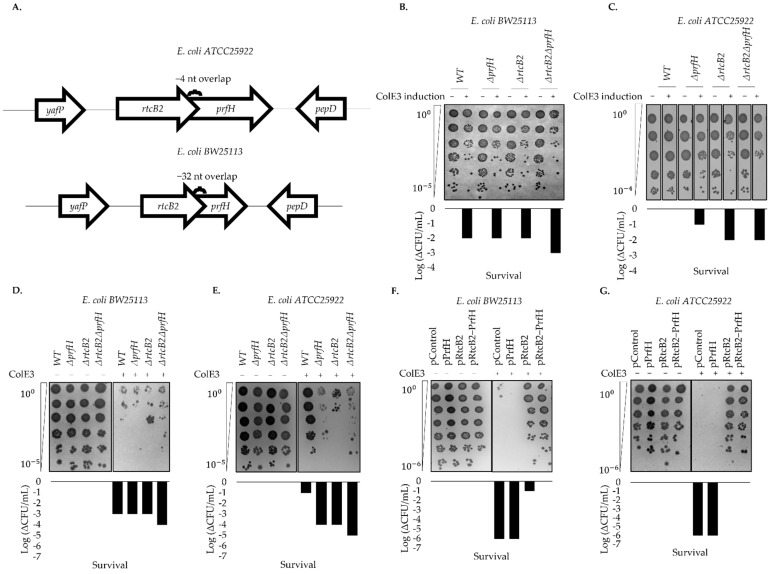
Action of colicin E3 on different *E. coli* strains. (**A**) Location of *rtcB2* and *prfH* genes in the genome of *E. coli* strains used in the current study. The top and bottom regions show operon regions for *E. coli ATCC25922* and *BW25113*, respectively. Survival upon colicin E3 treatment is shown from (**B**–**G**). The results from (**B**–**G**) demonstrate a decline of cell titer after serial dilutions. (**B**,**C**) show in vivo testing of colicin E3 activity on wild-type and knockout strains after inducible expression of colicin E3 gene under tet promoter, executed for both *E. coli BW25113* and *E. coli ATCC25922* strains. Colicin E3 synthesis was induced with 0.2 µg/mL of anhydrotetracycline when cells had grown to log-phase. (**D**,**E**) are an in vivo test for the recombinant colicin E3 activity on wild-type and knockout strains for *E. coli BW25113* and *E. coli ATCC25922*, respectively. (**F**,**G**) is a demonstration of survival assay in *rtcB2*-*prfH* double knockout cells, transformed by complementation plasmids in both *E. coli BW25113* and *E. coli ATCC25922* strains. For (**D**–**G**) the cleavage effect was induced upon treatment with 1 mg/mL of recombinant colicin E3 when cells reached log-phase. Milli-Q ultrapure water was used as a negative control for the experiment. Together, all these results were consistent and reproducible on repeat at least twice. Wherein a number of countable colonies (>10 colonies) from the most serially diluted spot scored in the survival assay, the order of magnitude difference was used to determine the decrease in cell growth between treated and corresponding untreated samples, respectively.

**Figure 2 ijms-23-06453-f002:**
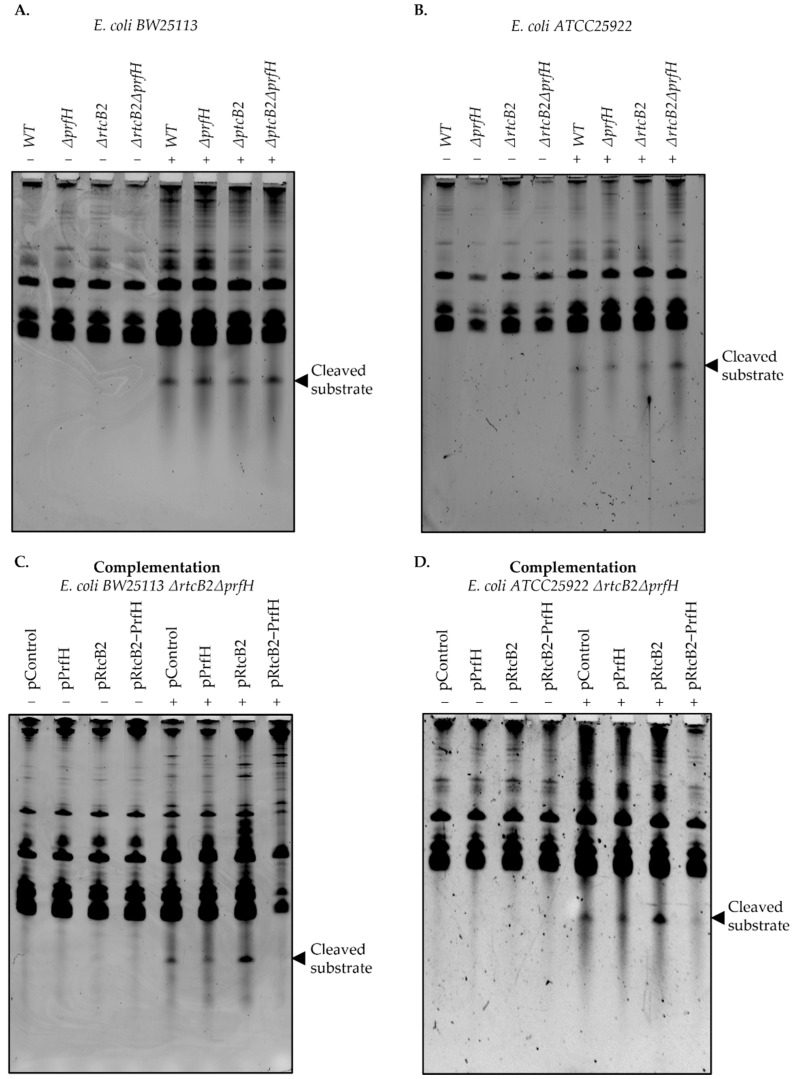
In vivo rRNA cleavage upon colicin E3 treatment. Electrophoretic separation of the in vivo 16S rRNA colicin E3 cleavage fragments for wild-type and knockout strains shown in (**A**,**B**), including complementation (**C**,**D**) of the double knockouts genotypes bearing single or both, *rtcB2*-*prfH*, gene(s). Bacterial cells untreated/treated with colicin E3 samples were used to isolate total RNA, and the resulting denaturing polyacrylamide gel electrophoresis was used to analyze rRNA fragments generated by the rRNAse. The − and + signs mean untreated and treated samples with either dialysis buffer (Milli-Q ultrapure water) or 1 mg/mL of recombinant colicin E3 protein, respectively. The results were consistent and reproducible on repeat at least twice upon treatment with colicin E3.

**Figure 3 ijms-23-06453-f003:**
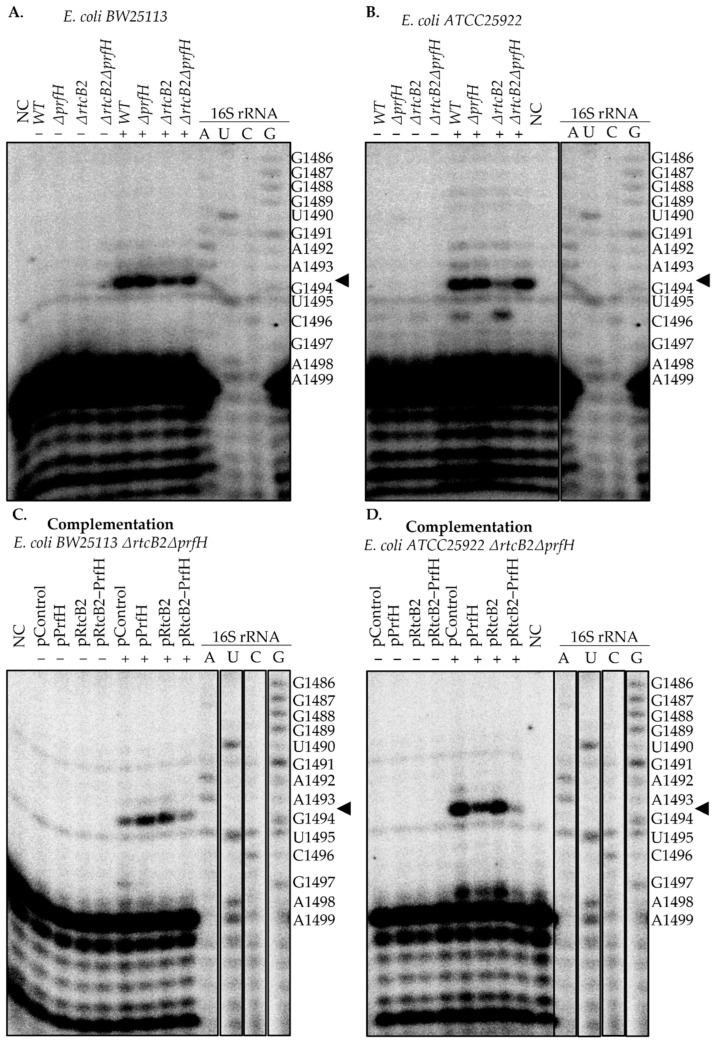
Primer extension results showing the cleavage analysis of 16S rRNA under in vivo colicin E3 treatment for wild-type and knockout strains shown for (**A**,**B**), including complementation (**C**,**D**) of the double knockouts with single and both gene expression from the *rtcB2*-*prfH* operon. Untreated and treated bacterial cells with colicin E3 were used to isolate total RNA. About ~500 ng of template RNA was used for annealing with a γ-[^32^P] ATP labeled reverse primer complementary to the 16S rRNA. The resultant cDNA products synthesized by AMV reverse transcriptase were then separated by a 20% denaturing polyacrylamide gel electrophoresis. The sequence reactions were used to map the cleavage sites obtained. The results were consistent and reproducible at least twice when the experiment was repeated under treatment with colicin E3. NC is a negative control, wherein Milli-Q ultrapure water was used instead of total RNA purified from the samples.

**Figure 4 ijms-23-06453-f004:**
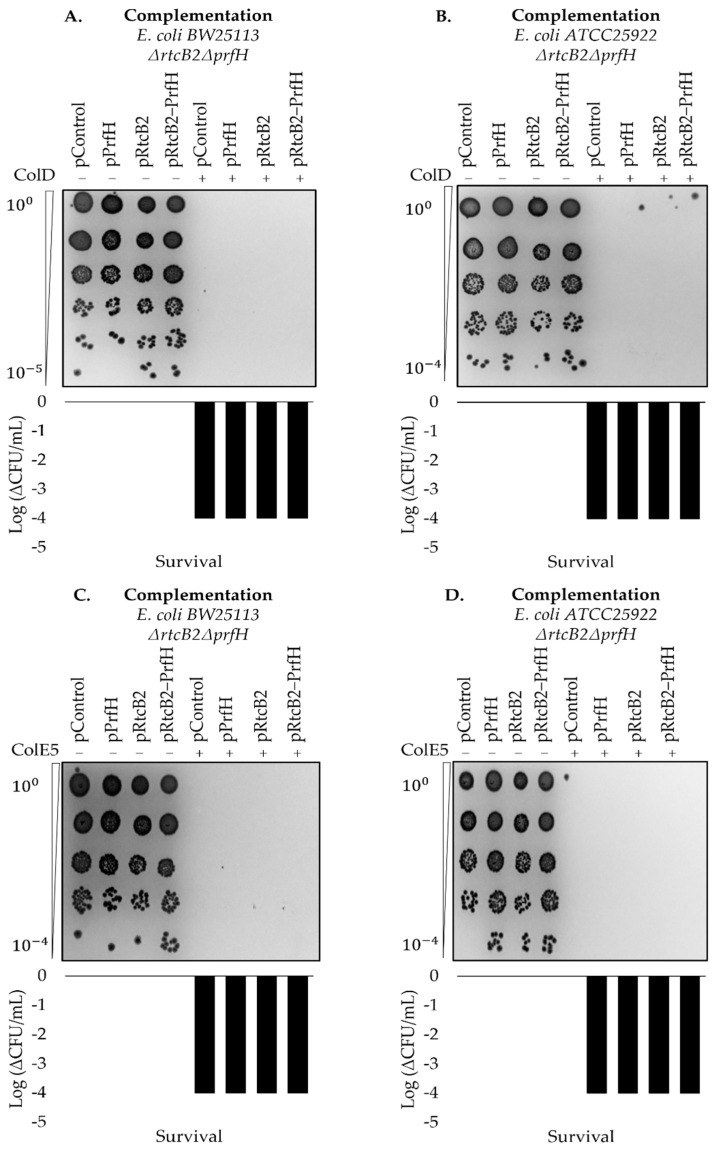
Survival upon colicin E5 and D treatment. The results demonstrate a decline of cell titer after serial dilutions. This is a demonstration of survival assay(s) in *rtcB2*-*prfH* double knockout cells, transformed by complementation plasmids in both *E. coli BW25113* and *E. coli ATCC25922* strains treated with recombinant colicin D (for (**A**,**B**)) and E5 (for (**C**,**D**)) at log-phase, respectively. The results were consistent and reproducible on repeat at least twice.

## Data Availability

Not applicable.

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
