# Peer review of "RtcB2-PrfH Operon Protects E. coli ATCC25922 Strain from Colicin E3 Toxin"

_ijms, 2022, doi:10.3390/ijms23126453_

Round 1

Reviewer 1 Report

The manuscript by Maviza et al. presents results on the RtcB2-PrfH system to protect E. coli from colicin E3, a ribotoxin. The authors use in vivo studies to demonstrate the importance of this system in protecting (survival) the E. coli strains from colicin E3. The results are nicely presented and conclusions drawn are supported by the data presented. The following comments should be addressed.

(i) Page 2, lines 68-80: the authors describe recent work that colicin E3 damaged ribosomes are substrates for PrfH but do not provide a reference. Please provide the reference. Only reference 18 is used in these sentences, which is neither recent (2006) and relevant to PrfH activity, i.e., this reference reports on colicin E5.

(ii) There are a number of seminal references missing on characterization of RtcB2. Please include these in the introduction. For example:

Banerjee A, Goldgur Y, Shuman S. RNA. 2021 Feb 22;27(5):584-90.

Tanaka N, Shuman S. J Biol Chem. 2011 Mar 11;286(10):7727-7731.

Tanaka N, Meineke B, Shuman S. J Biol Chem. 2011 Sep 2;286(35):30253-30257.

Tanaka N, Chakravarty AK, Maughan B, Shuman S. J Biol Chem. 2011 Dec 16;286(50):43134-43.

Englert M, Xia S, Okada C, Nakamura A, Tanavde V, Yao M, Eom SH, Konigsberg WH, Söll D, Wang J. Proc Natl Acad Sci U S A. 2012 Sep 18;109(38):15235-40.

(iii) Figure 3: Why do the cleaved products in 3C differ to those in panels A, B and D of Figure 3, i.e., different electrophoresis pattern and migration? Include size markers in the gels presented in Figure 3.

(iv) Figure 5: The authors use crude supernatants that apparently contain secreted colicin E5 and D toxins. No evidence is provided that the supernatants contain these toxins following preparation. This information documenting the presence of these colicins should be provided as Supplementary data.

(v) In text references 25 and 26 are missing in the reference list.

Reviewer 2 Report

In this manuscript, the authors demonstrated that RtcB2-PrfH operon could protect E coli., ATCC25992 strain in particularly, from ribotoxicity caused by Colicin E3 toxin. The protection effect is dependent on bacteria strains and the types of damage induced by different toxins. The experiments were designed and performed well, the manuscript was also organized well. The Method section was described clearly as well. The concept itself is interesting and the findings may benefit the field. However, the molecular functions of RtcB and PrfH (even the cyro-EM structure of PrfH-ribosome complex) have been demonstrated by a recent publication in bioRxiv (Ref 16). In the current manuscript, the authors performed a certain degree of “validations” in bacterial cells (BTW, I do not believe the cell-based analyses can be called as “in vivo”), so that the novelty of the main findings here is largely limited. Also, as the authors discussed in the Discussion section (start from line 218), the loss-of-function and gain-of-function results of PrfH were not consistent very well. Indeed, the expression level could be an important contributing factor. However, there is no direct evidence demonstrating the protein expression levels in different stains (for both WT, knockout and over-expression strains). So that, it is not suitable to make a conclusion based on current dataset. Couple of minor issues:

  1. In the Results part, the authors mentioned Figure 2 earlier than Figure 1. In this case, the authors should modify the text or move Figure 1 backward.
  2. The survival assay is good but not quantitative. Could you somehow quantify the toxicity of Colicin E3 toxin in different stains?
  3. Language needs to be edited.

Round 2

Reviewer 2 Report

The authors partially answered my concerns. I cannot see quantification except new Fig 1. The authors did not comment on the novelty issue and the inconsistent results between loss-of-function and gain-of-function of PrfH.

Author Response

  1. The authors partially answered my concerns. I cannot see quantification except new Fig 1.

Thank you for the prompt feedback. We have quantified all our Figures, in addition, accordingly, now presented as: Figure 4 (added bar graphs on Survival assays) Figure S4 and S5 (quantified cleavage fragment zones using ImageLab software and generated a bar graph on the corresponding band or lane respectively).

  1. The authors did not comment on the novelty issue and the inconsistent results between loss-of-function and gain-of-function of PrfH

Thank you for kindly reminding us on this issue.

With respect to the “inconsistent results between loss-of-function and gain-of-function of PrfH”:

  • We paraphrase our answer as follows: -

The protective effect aided by rtcB2 gene complementation in our double-knockout genotype, renders cells resistant to colicin E3 effect, with respect to our survival assays. This makes us speculate that prfH can be dispensable, when the rtcB2 gene is expressed at high-levels alone. Colicin E3 may cleave the 30S small subunit prior to the formation of a competent 70S ribosome at initiation stage, which may necessitate the need for repair by RtcB2, an independent event that does not require PrfH. In addition, alternative ribosome disassembly factors may aid in alleviating the colicin E3 damaged ribosomes when PrfH is absent, which still warrants further experimental validation which is beyond the scope of the current study.

We have modified this information to our discussion part to balance our findings from our study.

With respect to the “comment on novelty issue”:

  • We paraphrased our answer as follows: - The current study is an extension from a previous in vitro demonstration of the RtcB2-PrfH module in rescue and repair of colicin E3 damaged ribosome. We further complement the in vitro study using a cell-based system, wherein we demonstrate the indispensable effect exerted by this module, as an essential feature for cell viability under stress by colicin E3. We further show that the expression of the rtcB2 gene alone seem to render resistance to the ribotoxicity caused by colicin E3, which still warrants further investigation, on its possible stage of reparation of cleaved 16S rRNA, either alone or upon assistance by other factors, when PrfH is absent.

We have modified our conclusion and placed this part in support of our novelty in this current study.